# Strabismus Detection in Monocular Eye Images for Telemedicine Applications

**DOI:** 10.3390/jimaging10110284

**Published:** 2024-11-07

**Authors:** Wattanapong Kurdthongmee, Lunla Udomvej, Arsanchai Sukkuea, Piyadhida Kurdthongmee, Chitchanok Sangeamwong, Chayanid Chanakarn

**Affiliations:** 1School of Engineering and Technology, Walailak University, Thai Buri, Thasala, Nakornsithammarat 80160, Thailand; 2School of Medicine, Walailak University, Thai Buri, Thasala, Nakornsithammarat 80160, Thailand; 3The Center for Scientific and Technological Equipment, Walailak University, Thai Buri Thasala, Nakornsithammarat 80160, Thailand

**Keywords:** strabismus, ocular misalignment, early detection, telemedicine, screening

## Abstract

This study presents a novel method for the early detection of strabismus, a common eye misalignment disorder, with an emphasis on its application in telemedicine. The technique leverages synchronized eye movements to estimate the pupil location of one eye based on the other, achieving close alignment in non-strabismic cases. Regression models for each eye are developed using advanced machine learning algorithms, and significant discrepancies between estimated and actual pupil positions indicate the presence of strabismus. This approach provides a non-invasive, efficient solution for early detection and bridges the gap between basic research and clinical care by offering an accessible, machine learning-based tool that facilitates timely intervention and improved outcomes in diverse healthcare settings. The potential for pediatric screening is discussed as a possible direction for future research.

## 1. Introduction

Strabismus, a common vision disorder where the eyes are misaligned, is a significant public health concern, especially if left undetected and untreated. Early identification of strabismus is crucial to prevent severe consequences such as amblyopia (lazy eye) and reduced quality of life. While traditional clinical assessments by trained healthcare professionals are still the gold standard for diagnosing strabismus, recent technological advancements have introduced a new era of automated screening methods.

In the dynamic field of strabismus screening, numerous methodologies have evolved over the years to address the challenges of detecting ocular misalignment. Traditional approaches to strabismus screening have predominantly relied on manual assessments conducted by skilled ophthalmologists. These assessments involve a battery of tests, including the cover and uncover test, prism cover test, and the Hirschberg test. While these methods have been instrumental in identifying strabismus, they are often labor intensive, time consuming, and subject to variations in interpretation.

Recognizing the limitations of traditional screening methods, endeavors have been made to integrate digital tools and cutting-edge technologies into the strabismus screening landscape. Among these innovations, photoscreeners have emerged as promising tools, offering the potential to streamline and enhance the efficiency of strabismus screening [1,2,3,4]. These devices capture high-resolution images of the eyes, facilitating the detection of strabismus. Eye-tracking systems, particularly prevalent in larger settings such as schools, have been leveraged to identify strabismus with precision and objectivity [5,6]. These systems meticulously monitor eye movements and provide valuable data for diagnosis. Additionally, virtual reality headsets, equipped with pupil-tracking technology, have entered the arena, enabling the precise measurement of ocular deviation for strabismus screening [7,8,9]. However, it is noteworthy that these technologies often come with a substantial price tag, potentially limiting their accessibility in regions with restricted access to advanced medical resources.

In recent years, there has been a surging interest in harnessing digital images and automated techniques to enhance the efficiency and scalability of strabismus screening [10,11,12]. For instance, Almeida et al. [10] achieved successful strabismus detection by analyzing digital images acquired during the Hirschberg test, while Sousa de Almeida et al. [13] utilized images capturing gaze positions to identify strabismus. Valente and collaborators [12] incorporated images from the cover test into their screening methodology. Moreover, Sousa de Almeida et al. [13] applied images capturing gaze positions for strabismus identification, further emphasizing the potential of digital image-based approaches in strabismus detection. These studies collectively emphasize the effectiveness of leveraging digital images from clinical tests for strabismus detection and screening [3].

Deep learning techniques have made significant strides in the domain of strabismus detection, demonstrating their potential to automate the diagnostic process with an impressive degree of precision [11,14]. However, it is essential to highlight some distinctions between our approach and these deep learning methods. While the deep learning techniques mentioned have excelled in achieving exceptional classification accuracy, they often yield results that can be challenging to interpret due to the intricate internal processes of deep neural networks [3]. In contrast, our approach distinguishes itself by offering a more interpretable methodology, driven by the combination of direct and cross-eye regression models. This unique approach simplifies the output, providing more intuitive and clinically relevant information for practitioners and enhancing the ease of decision making in strabismus diagnosis. Additionally, our method eliminates the need for precise measurements and strict subject positioning, making it versatile for both normal and strabismic eyes, thus addressing potential limitations associated with other deep learning-based approaches. These distinctions position our approach as a promising candidate for strabismus screening, combining the benefits of deep learning with improved interpretability and practical applicability.

This paper introduces a comprehensive review of the latest research and developments in the automation of strabismus screening, highlighting the contributions and advancements in the field. It presents an innovative approach to strabismus detection that combines both direct and cross-eye regression models, enhancing the accuracy and reliability of strabismus screening. Additionally, the methodology leverages advanced machine learning techniques, creating regression models for each eye and conducting benchmarking comparisons with existing strabismus detection methods. The proposed approach eliminates the need for precise measurements and strict subject positioning, making it versatile for both normal and strabismic eyes. By offering a cost-effective and scalable solution, this approach has the potential to improve access to strabismus screening, even in resource-limited settings. It extends the reach of early diagnosis and intervention, reducing the burden of strabismus-related visual impairments.

The structure of this paper can be outlined as follows: Section 2 elaborates on the materials and methods employed. Section 3 presents the experimental results and subsequent discussions. Lastly, Section 4 provides the paper’s concluding remarks.

## 2. Materials and Methods

In the context of this paper, we introduce and define two critical terms that form the foundation of our approach to strabismus detection:Cross-Eye Regression: Cross-eye regression is a machine learning technique employed to estimate the pupil’s location in one eye based on image data from the opposite eye. In simpler terms, it predicts the pupil’s position in one eye, such as the right eye, using image information from the contralateral eye, such as the left eye. This approach capitalizes on the natural correlation between the movements of both eyes during gaze. In individuals without strabismus, this estimation closely aligns with the actual pupil location, facilitating early strabismus detection.Direct Eye Regression: Conversely, direct eye regression involves the direct prediction of the pupil’s location within an individual eye image. Rather than relying on information from the opposite eye, this approach aims to predict the pupil’s position within the same eye image. It proves particularly valuable for assessing the alignment and gaze of each eye independently. In our research, we leverage advanced machine learning techniques to develop models that directly forecast pupil locations, thereby enhancing the accuracy and efficiency of strabismus detection.

These two techniques form the basis of our methodology for early strabismus detection, where cross-eye regression and direct eye regression play distinct but complementary roles in our model’s ability to diagnose and assess this common ocular misalignment condition.

### 2.1. Datasets

In this research, we harnessed a diverse array of datasets to develop and assess our strabismus detection model, establishing the cornerstone for training, testing, validation, and benchmarking our approach. Notably, we employed the following pivotal datasets, each contributing distinct facets to our study:Columbia Gaze Dataset [15]: The Columbia Gaze Dataset serves as a foundational resource for our research, facilitating the training of strabismus detection models. This dataset comprises a diverse collection of eye images, encompassing a wide range of gaze directions and head orientations from 56 subjects, each participating in sessions with 21 fixed gaze targets across five head orientations. Although it primarily includes adult subjects and lacks detailed age information, it provides valuable insights into general gaze behavior. However, this adult focus limits its applicability for pediatric strabismus screening, which is essential for effective early diagnosis. To enhance the model’s relevance for children, future research should prioritize the collection of pediatric-specific data. For each image, we meticulously created annotation files with precise coordinates for the left and right pupil locations, crucial for model training and evaluation. The high resolution of the images (5184 × 3456 pixels) enhances the dataset’s quality, contributing to the accurate prediction of pupil positions. Nonetheless, the limited age diversity in the dataset is a significant limitation that future studies must address to broaden its applicability in pediatric strabismus detection.GI4E Dataset [16]: The GI4E dataset, housing 1236 images in PNG format with a resolution of 800 × 600 pixels, plays a crucial role as a benchmark in our research. This dataset is instrumental in model validation and fine tuning. Although it does not include age data, it serves as a valuable resource for experimentation with diverse loss functions, training epochs, and optimizers, facilitating the optimization of our strabismus detection models.Video Datasets from Dreamstime (www.dreamstime.com, accessed on 16 June 2024): To scrutinize our model’s real-world applicability and dynamic performance comprehensively, we incorporated video datasets sourced from Dreamstime. These video datasets encompass recorded clips featuring individuals with strabismus, capturing a broad spectrum of eye movements and conditions. All videos conform to a standardized resolution of 489 × 870 pixels and are encoded in MP4 format. These datasets furnish valuable insights into strabismus detection within practical scenarios, providing dynamic and real-time perspectives. Detailed information concerning these video datasets is presented in Table 1.

The fusion of these datasets underscores the versatility, precision, and adaptability of our strabismus detection methodology, yielding a robust solution for early strabismus diagnosis across diverse scenarios. However, it is important to note that the absence of age-specific data, particularly for younger children, remains a limitation in the current datasets. Future efforts will be directed toward incorporating pediatric-focused datasets to address this gap and improve the utility of the system in early childhood strabismus screening.

### 2.2. Methods

In our proposed methodology, we have harnessed advanced machine learning techniques to establish an innovative approach for strabismus detection. This approach is rooted in two fundamental principles we introduced earlier: “cross-eye regression” and “direct eye regression”. These principles capitalize on the intrinsic relationship between the movements of both eyes during gaze, enabling us to predict pupil positions accurately and enhance the accuracy of strabismus detection.

Cross-eye regression leverages the natural synchronization of eye movements in individuals without strabismus. We meticulously design and train two separate regression models, each dedicated to one eye. These models are fortified with the Xception (Extreme Inception) backbone architecture [17], enabling them to process an image of one eye and generate precise pupil location predictions for the corresponding eye on the opposite side. For instance, inputting an image of the left eye empowers us to forecast the pupil location of the right eye accurately, providing a novel perspective on eye movement coordination.

Simultaneously, our methodology encompasses direct eye regression. Here, we develop an independent regression model that directly predicts the pupil location of the eye under examination. This versatile approach allows us to explore eye movement coordination within each eye individually.

The Xception architecture is a powerful deep convolutional neural network (CNN) known for its effectiveness in various computer vision tasks, including image classification and feature extraction. However, adapting Xception for regression tasks, such as predicting pupil locations in strabismus detection, requires significant changes to its architecture, particularly in the final layers.

Typically, the terminal layers of the Xception model are designed for classification, using activation functions like softmax to assign discrete labels. In contrast, for regression tasks that aim to predict continuous coordinates (x,y), the architecture undergoes a substantial modification. As shown in Figure 1, the original classification layers are replaced with dense layers specifically tailored to capture the intricate patterns and relationships within the extracted features.

Additionally, the activation function in the output layer transitions from a classification-oriented function to a linear one. This change is crucial, as it enables the network to produce continuous, real-valued outputs—the exact coordinates of the pupil’s position.

The journey towards optimizing our regression model for strabismus detection unfolds through a meticulously planned series of experiments. Each experiment represents a strategic exploration of various facets critical to our model’s performance. We systematically evaluate a range of loss functions, carefully selecting those that steer our model towards the utmost accuracy in predicting pupil locations. To gauge the model’s effectiveness, we rely on key metrics like the mean absolute error (MAE) and root mean squared error (RMSE), which serve as guiding benchmarks throughout our iterative refinement process. Furthermore, we fine tune the number of epochs, ensuring precise calibration that strikes a delicate balance between model convergence and the prevention of overfitting. Additionally, we explore the impact of varying the size of the input image to the Xception model, examining its influence on the overall performance of our strabismus detection system.

In this manner, our approach strives for excellence, fusing the robust feature extraction capabilities of the pre-trained Xception model with a customized and finely tuned regression head. The result is a powerful strabismus detection tool, honed through meticulous experimentation, poised to predict pupil locations with exceptional accuracy.

In our pursuit of accurate strabismus detection, we utilize Google’s MediaPipe, a robust computer vision framework renowned for its versatile facial feature detection capabilities. MediaPipe holds a central position in our methodology as it enables the precise extraction of eye regions from input images, serving as the cornerstone for subsequent steps, including cross-eye and direct eye regression. This seamless integration significantly enhances the accuracy of our strabismus detection methodology. We meticulously configure its parameters to align with our specific research objectives, ultimately bolstering the reliability of our outcomes. Importantly, we rely on our own direct eye regression rather than MediaPipe’s pupil locations to overcome potential deviations in pupil position accuracy.

Figure 2 presents a comprehensive overview of the steps required to create the final direct eye regression model. This model is trained using the GI4E dataset, which incorporates annotation data. To facilitate the generation of essential datasets (training, testing, and validation), a dedicated Python script (version 3.9.2) was developed, allocating data in an 80:10:10 ratio. The first two datasets were employed for training and evaluating the Xception backbone with a regression head. Multiple models were generated, exploring various combinations of epochs, optimizers, and regression functions. The selection of the optimal direct eye regression model was determined through validation using the validation dataset. Our custom Python script calculated the mean squared error (MSE) against the ground truth, ultimately identifying the model with the lowest MSE as the final direct eye regression model.

The process of creating the cross-eye regression model commences with the preparation of cross-eye datasets, specifically the left eye image alongside the estimated location of the right pupil. These steps are elucidated in Figure 3. The Columbia Gaze Dataset serves as the source, and MediaPipe is harnessed to extract regions of both eyes. Only eyes with approximately equal sizes, falling within a size difference range of 0.90 to 1.10, are deemed acceptable. This criterion ensures that the cross-eye pupil location remains within a similar scale as the eye image. Subsequently, the direct eye regression model is applied to these eye regions to estimate pupil locations. It is important to note that for each image containing a pair of detected eyes, this step yields two distinct outputs: (1) the left eye image alongside the estimated location of the right pupil (resulting from applying the direct eye regression model to the right eye image), and (2) the right eye image alongside the estimated location of the left pupil.

The steps involved in creating the final cross-eye regression model closely parallel those for the direct eye regression model, as detailed in Figure 2. The key difference lies in the input datasets, which are substituted with the ‘Right Eye Image with Left Estimated Pupil dataset’ for the ultimate cross-eye regression model of the right eye image. Likewise, for the final cross-eye regression model of the left eye image, the input datasets are changed from ‘GI4E images and annotations’ to ‘Left Eye Image with Right Estimated Pupil dataset’.

To establish the robustness of our methodology, we conduct rigorous statistical analyses using data from individuals free of strabismus. These analyses yield crucial threshold values indicative of typical eye movement correlation—a vital foundation for our approach.

Subsequently, we validate our approach through experiments involving individuals diagnosed with strabismus. The comprehensive statistical analysis of these experiments reveals significant disparities between the “normal” and “strabismus” groups. These findings not only affirm the effectiveness of our approach but also underscore its practical utility for early strabismus detection, whether through cross-eye or direct eye regression.

### 2.3. Performance Evaluation Metrics

To rigorously assess the accuracy of our strabismus detection models, we employed multiple key performance metrics, tailored to different aspects of the detection process. Specifically, we utilized *MSE* for both cross-eye and direct eye regression models, in addition to the Euclidean distance (*D*) for classifying whether an individual has normal or strabismus eyes.

*MSE* measures the average squared differences between the actual and predicted pupil positions. It serves as a critical indicator of model accuracy, especially during the regression process. The *MSE* for each eye is calculated as follows:(1)$$MSE=\frac{1}{n}\overset{n}{\underset{i=1}{\sum }}{\left(P_{i}-{\hat{P}}_{i}\right)}^{2}$$
where *MSE* represents the Mean Squared Error for the eye. *n* is the total number of samples. $P_{i}$ denotes the actual pupil position for a given sample. ${\hat{P}}_{i}$ signifies the predicted pupil position for the same sample.

In addition to *MSE*, we employed the Euclidean distance as a pivotal metric for classification purposes. The Euclidean distance quantifies the straight-line separation between the pupil positions estimated by the cross-eye and direct eye regression models within each eye. It provides a quantitative gauge of how closely the actual pupil positions align with the expected pupil positions of both eyes during gaze.

The Euclidean distance, *D*, is calculated using the following equation:(2)$$D=\sqrt{{\left(x_{2}-x_{1}\right)}^{2}+{\left(y_{2}-y_{1}\right)}^{2}}$$
where *D* represents the Euclidean distance between two points, which is used to measure the discrepancy between the estimated pupil positions. The coordinates $\left(x_{1},y_{1}\right)$ are obtained from the cross-eye regression model, while $\left(x_{2},y_{2}\right)$ are derived from the direct eye regression model. These coordinates are taken from separate images of the left and right eyes, respectively, not from a single face image. The calculation of the Euclidean distance *D* focuses on the relative difference between the estimated positions of the pupils, making it independent of the frame choice. This ensures that the measure accurately reflects the difference in pupil positions across both images.

These metrics collectively provide a comprehensive evaluation of our strabismus detection models, encompassing both the regression accuracy and the ability to classify normal and strabismus eyes.

For each eye, we calculated the Euclidean distances for all frames in the dataset. This process enabled us to derive essential statistical metrics, including the mean (*µ*), standard deviation (*σ*), minimum (min), and maximum (max) distances.

### 2.4. Classification Threshold for Strabismus Detection

To discern whether an individual exhibits strabismus or not, we apply a classification threshold based on the Euclidean distances of pupils from the normal dataset. For each eye, we compute the threshold ($\Theta_{left}$ and $\Theta_{right}$) using the following formulas:(3)$$\Theta_{left}=\mu_{left}\pm K\cdot \sigma_{left}$$
(4)$$\Theta_{right}=\mu_{right}\pm L\cdot \sigma_{right}$$

The parameters *K* and *L* in Equations (3) and (4) represent threshold values for the left and right eyes, respectively, and they may vary depending on the dataset used. These thresholds are critical in determining whether the estimated pupil positions exceed acceptable limits for normal alignment. For any given test image, if the Euclidean distance for either the left or right eye surpasses its respective threshold (*K* or *L*), the individual is classified as having strabismus. This approach ensures that both eyes are independently assessed, providing a more comprehensive evaluation of potential strabismus.

### 2.5. Methodology Overview

Figure 4 provides an overview of our strabismus detection methodology. The process begins with input eye region bounding boxes obtained from MediaPipe (https://pypi.org/project/mediapipe/, accessed on 27 October 2024). To ensure consistency, the bounding boxes for both eyes are required to have comparable sizes within a permissible range of 0.9 to 1.1. The following steps are then applied to the input eyes.
Cross-Eye Regression: Using both eye bounding boxes as input, our cross-eye regression model estimates the positions of the pupils. These estimated pupil positions are represented by red plus signs at the middle top of the figure.Direct Eye Regression: Similarly, we employ direct eye regression, again using both eye bounding boxes as input, to estimate the positions of the pupils. These estimated pupil positions are depicted as green plus signs at the middle bottom of the figure.

For each eye, we compute the Euclidean distance (*D*) between these two estimated pupil positions. These distances are then compared to the threshold (Θ) for classification. If the distances fall within the expected range, the classification result is “normal eye”. Conversely, if the distances exceed the threshold, the classification result is “strabismus”.

## 3. Results and Discussion

In this section, we present the detailed results of our experiments, covering both cross-eye and direct eye regression models. These models were thoroughly evaluated across different configurations and training periods, specifically focusing on training epochs between 300 and 350. We tested various input image sizes, ranging from 72 × 72 to 196 × 196 pixels, to understand their impact on model performance and reliability. Among these, an input size of 128 × 128 pixels consistently provided the best balance of accuracy and computational efficiency. While we did not include 64 × 64 and 256 × 256 pixel sizes in this study, we acknowledge that they could also offer useful insights and will consider them for future evaluations.

To ensure a comprehensive assessment, we used *MSE* as the main metric to measure both the accuracy and reliability of our models in test scenarios and real-world video clips. It is important to note that normal and strabismus cases were classified based on the Euclidean distance, which measures the difference between pupil positions predicted by the cross-eye and direct eye regression models. This distance serves as a quantitative measure of how closely the predicted pupil positions align with the actual positions during gaze.

### 3.1. Cross-Eye and Direct Eye Regression Models

In this section, we delve into the results of our cross-eye and direct eye regression model experiments, with a specific emphasis on training epochs ranging between 300 and 350. These experiments aimed to estimate the pupil’s location in one eye based on image information from either the same eye (direct eye) or the opposite eye (cross-eye). We explored a range of model configurations and training epochs, with the primary evaluation metric being *MSE*. Additionally, we emphasize the crucial role of an input feature size of 128 × 128 pixels, which consistently yielded optimal results.

### 3.2. Model Configurations and Epoch Selection

The selection of training epochs between 300 and 350, combined with the RMSprop optimizer and MSE evaluation, was guided by a meticulous rationale. Our objective was to identify the training conditions that consistently provided accurate and stable predictions for pupil locations. This involved an exhaustive exploration of different model configurations, including variations in deep learning models, optimization algorithms, and loss functions.

The choice of RMSprop as the optimizer stemmed from its adaptive learning rate mechanism, facilitating smoother convergence during training. This adaptability proved advantageous for intricate tasks like pupil location estimation, where convergence challenges can arise due to variations in eye images and gaze conditions. Our systematic experimentation with alternative configurations and epochs revealed a consistent trend: configurations other than those within the specified epoch range, or employing different optimizers and loss functions, resulted in notably higher *MSE*s. These higher *MSE* values indicated less accurate predictions and greater variance in pupil location estimations.

### 3.3. Top Cross-Eye and Direct Eye Regression Models

Table 2 offers a comprehensive summary of the *MSE* results obtained within the specified epoch range for both cross-eye and direct eye regression models. The table encompasses *MSE* values and includes critical statistical metrics like standard deviation (SD), minimum (Min), and maximum (Max) values of the squared errors for both the left and right eyes. These metrics provide valuable insights into the stability and consistency of the models.

In the context of direct eye regression models, we present the results of the top-performing five configurations, as summarized in Table 3. These models were trained over 250 epochs and assessed primarily using MSE as the evaluation metric.

Among these models, Nadam with MAE achieved an *MSE* of 5.37, with an SD of 2.26. The model’s squared error ranged from a minimum of 0.49 to a maximum of 12.03.

Adam with Log Cosh loss resulted in an *MSE* of 5.69, accompanied by an SD of 2.22. The model exhibited a minimum squared error of 0.18 and a maximum squared error of 12.81.

Adam with MAE produced an *MSE* of 5.82, with an SD of 2.35. The model’s squared error values ranged from a minimum of 0.04 to a maximum of 12.64.

Nadam with MSE loss function yielded an *MSE* of 6.03, along with an SD of 2.09. The model’s minimum and maximum squared error values were 0.42 and 12.12, respectively.

Adam with Mean Absolute Percentage Error achieved an *MSE* of 6.18, accompanied by an SD of 2.09. The model’s squared error ranged from a minimum of 0.55 to a maximum of 12.70.

These top direct eye regression models represent configurations that excel in minimizing *MSE* and providing consistent estimates of pupil locations. The metrics offer valuable insights into the accuracy and stability of these models, guiding the selection of the most suitable direct eye regression model for subsequent strabismus detection experiments.

### 3.4. Conclusion of Cross-Eye and Direct Eye Regression Models

After a rigorous evaluation process, we have successfully identified the best-performing models among those studied for our subsequent experiments. For cross-eye regression models, we have chosen the configuration that utilizes Xception with the RMSprop optimizer and MAE loss function, trained over 325 epochs. Conversely, for direct eye regression models, we have determined that employing Nadam with the MAE loss function, trained for 250 epochs, consistently yields exceptional results. Furthermore, our investigations have underscored the significance of an input feature size set at 128 × 128 pixels, consistently producing the most favorable outcomes. These findings establish a robust foundation for our forthcoming strabismus detection experiments, guaranteeing the precision and reliability of pupil location estimation.

### 3.5. Benchmarking Results

In our study, we conducted a benchmark comparison between our proposed approach and the method developed by Huang et al. [3] using a dataset generously provided under a Creative Commons BY 4.0 license. This dataset consisted of 30 samples of normal eyes and 30 samples of eyes affected by strabismus. Importantly, this dataset contained complete eye regions, rendering the utilization of MediaPipe for eye region extraction unnecessary. However, it is worth noting that the images in this dataset were grayscale, with some annotations applied as part of Huang et al.’s methodology. This grayscale format and the presence of annotations in the dataset have the potential to introduce challenges that could adversely affect the performance of our proposed approach. It is important to acknowledge that the availability of suitable benchmark datasets for strabismus detection remains limited, with many proprietary datasets posing challenges to researchers aiming to conduct comprehensive benchmarking studies.

In Figure 5, we present the box plot results obtained from our extensive statistical analysis of the dataset. These box plots offer valuable insights into the distribution of Euclidean distances for both left and right eyes among individuals with normal vision and those affected by strabismus. Notably, the box plots reveal distinct patterns in the data distribution. For the left eye, the box plot illustrates that the median Euclidean distance for individuals with strabismus is noticeably lower than that of individuals with normal vision, while exhibiting a wider interquartile range. Conversely, the right eye’s box plot showcases a higher median Euclidean distance for individuals with normal vision, along with a more concentrated distribution, in contrast to individuals with strabismus. These findings suggest that it is indeed possible to establish effective separation criteria between normal and strabismus cases based on the mean and standard deviation of Euclidean distances, as indicated by the thresholds Θ*_left_* and Θ*_right_*.

Building upon the promising results revealed by the box plots, we implemented a robust statistical approach to differentiate between normal and strabismus cases, with key parameters outlined in Table 4.

For the left eye, we computed a threshold, denoted as Θ*_left_*, by adding 1.5 times the standard deviation *σ_left_* to the mean *µ_left_* of the Euclidean distances:(5)$$\Theta_{left}=\mu_{left}+1.5\cdot \sigma_{left}$$

This threshold establishes a boundary beyond which data points are considered potential outliers, indicating the presence of strabismus. Similarly, for the right eye, the threshold Θ*_right_* is defined as follows:(6)$$\Theta_{right}=\mu_{right}+1.5\cdot \sigma_{right}$$

If the Euclidean distance for either eye exceeds its respective threshold—specifically, if the distance for the left eye, *D_left_*, is greater than the left threshold, Θ*_left_*, or if the distance for the right eye, *D_right_*, is greater than the right threshold, Θ*_right_*—then the case is classified as strabismus.

This approach effectively harnesses the central tendency (mean) and variability (standard deviation) of the data distribution, providing a reliable, data-driven method for identifying strabismus cases. Additionally, the threshold multiplier of 1.5 can be adjusted to fine tune the sensitivity and specificity of the detection process, accommodating different dataset characteristics and performance criteria.

The classification results are summarized in the last two columns of Table 4. Our approach demonstrates remarkable accuracy in strabismus detection, achieving a 100 percent correct classification rate. However, it is noteworthy that for the normal eye category, there was one image that was misclassified. To provide further insight, all images, along with their classification results, are presented in Figure 6 and Figure 7.

This exceptional case of incorrect classification, located in the last row of column 3, has prompted discussions with our medical advisor. It presents a unique challenge as it was classified as a normal eye by Huang et al. [3], while our medical advisor diagnosed it as strabismus, specifically esotropia. Notably, in all images classified as normal, the pupils, as estimated by both cross-eye and direct eye regression models, fall within the iris, which is the black region within both eyes. The anomaly in this particular image is the position of the right cross-eye pupil, which lies outside the iris.

Similar observations can be made for certain strabismus cases, especially those located in column 0 and 1 of the second row in Figure 7. These images also feature pupils positioned outside the expected range. In the case of the remaining strabismus cases, their classification is attributed to distances either greater than Θ*_left_* or beyond the range of Θ*_right_*, further underscoring the effectiveness of our approach in identifying strabismus based on these statistical thresholds.

### 3.6. Results on Video Datasets

Our experiments extended beyond benchmarking, encompassing the evaluation of our strabismus detection approach on real-world video clips. These video datasets included “normal” and “strabismus” categories, each containing six video clips featuring individuals with varying eye conditions. The primary objective of this evaluation was to assess the performance of our proposed approach in dynamic and uncontrolled environments, replicating real-world scenarios. This assessment aimed to determine the robustness and applicability of our methodology to practical situations, where factors like lighting, head movement, and variations in gaze direction can impact strabismus detection accuracy.

To ensure accurate assessments, we applied strict criteria for selecting frames. Specifically, we required that the sizes of the left and right eye regions be similar, within a defined ratio range of 0.9 to 1.1. This means that the area of one eye region could be up to 10% smaller or larger than the other. Frames that met this criterion were included in the analysis, while frames with a greater size difference between the eye regions were excluded to maintain consistency and reliability in the results.

We systematically analyzed every frame within the selected video clips, recording and aggregating the Euclidean distances generated by our approach. These distance data were subsequently visualized using box plots, as demonstrated in Figure 8.

The box plots provided valuable insights into the performance of our approach on video datasets. Notably, the results consistently grouped into two categories—“normal” and “strabismus”—aligning with the nature of the video clip datasets (detailed in Table 1).

Table 5 summarizes the statistics of our approach on video datasets. Building upon the observations from the box plots, we computed threshold values denoted as Θ*_left_* and Θ*_right_*. These thresholds were effectively determined by adding 1.5 times the standard deviation (σ) to the mean (µ) of the Euclidean distances. The statistics and thresholds are inserted between the “normal” and “strabismus” datasets in Table 5. They were placed there as they were calculated from the “normal” datasets. Following the same principles outlined in the benchmarking results section, the classification for each eye is defined as “normal” if the mean Euclidean distance is less than 6.21 and 9.64 for the left and right eye, respectively. The classification results for each eye and the final classification results are displayed in the last three columns of Table 5.

However, akin to the benchmarking phase, we encountered intriguing cases that deviated from the norm. Video clip 165165736 presented a compelling anomaly. Upon closer examination of this video clip, it initially exhibited characteristics of “normal” eye behavior. However, the individual intentionally manipulated their pupil, simulating strabismus only during the 00:03–00:05 timestamp. Remarkably, despite this deliberate act of mimicry, the model’s statistical analysis categorized this behavior as “normal”. This observation underscores the significance of context and intent in strabismus detection, suggesting that individuals may intentionally mimic or mitigate strabismus, thereby influencing the model’s assessments and outcomes.

In conclusion, our evaluation on real-world video datasets provides crucial insights into the robustness and adaptability of strabismus detection models. The observed patterns, influenced by the context and intent behind observed behaviors, underscore the importance of considering these factors when deploying such models in practical clinical settings or diagnostic applications. These results, coupled with our benchmarking findings, contribute to a comprehensive understanding of the capabilities and limitations of our approach.

### 3.7. Processing Time Results

The experiments for strabismus detection were conducted on a MacBook Pro M1 with 16 GB of memory, running the macOS Ventura operating system. The processing time per frame using our proposed approach was measured at 154.7 milliseconds (mS). This processing time corresponds to a frame rate of approximately 6.5 frames per second (FPS).

It is important to note that the benchmark approach used in this study did not provide processing time information. As a result, direct comparisons regarding processing time efficiency between our approach and the benchmark method are not feasible. However, the achieved frame rate of 6.5 FPS on a computer without a high-performance GPU demonstrates promising results. This level of performance indicates the potential applicability of our approach in telemedicine applications, where real-time or near-real-time strabismus screening can be of great significance.

### 3.8. Discussion

This study presents a novel approach to strabismus detection using cross-eye and direct eye regression models. Our models, built on advanced machine learning techniques, have demonstrated high accuracy in predicting pupil positions. While the findings suggest that our dual regression method has the potential to enhance early diagnosis and monitoring of strabismus, further clinical testing is necessary to validate its reliability as a clinical tool.

Current literature primarily relies on manual diagnostic methods or simpler automated techniques that lack robustness and accuracy. Our study addresses these gaps by introducing a high-precision detection system that leverages the correlation between eye movements. This dual approach not only improves detection accuracy but also offers a comprehensive analysis of eye alignment and movement.

Several studies have explored automated methods for detecting strabismus, but our approach distinguishes itself by integrating a dual regression model. Future research should include comparisons with emerging techniques, such as head-mounted displays and holographic augmented reality, to provide a more comprehensive evaluation of our method’s performance.

Despite the promising results, our study has limitations, particularly the absence of pediatric-specific data in the training set. The datasets used, especially the Columbia Gaze dataset, predominantly consist of adult subjects, limiting the model’s applicability for screening young children—an essential target population for early strabismus detection. Future work should prioritize collecting pediatric-specific data, particularly from children aged 1 to 10 years, to enhance the model’s relevance for early detection and intervention.

Additionally, our current model may face challenges in detecting conditions such as intermittent tropias, microtropia, and pupil abnormalities, which can affect accuracy. Unlike traditional screening methods that require subjects to maintain a fixed distance and gaze, our approach offers greater flexibility, making it suitable for uncooperative or younger subjects. This adaptability improves the likelihood of obtaining usable data during pediatric screenings.

While the flexibility of our system is an advantage, it may still lead to missed diagnoses in clinical settings, necessitating further research to quantify this impact. Addressing these limitations will require integrating advanced image processing techniques to account for variability in eye alignment and pupil behavior. Future versions could explore continuous video-based data collection to capture intermittent conditions and develop adaptive algorithms that function effectively with more challenging data.

The real-world applicability of our system was tested using video clips from Dreamstime, which may not fully represent the variability seen in clinical environments. Additionally, the high computational resources required for processing high-resolution images could limit practical deployment in some settings.

To overcome these limitations, future research should expand the dataset to include diverse eye conditions and demographics. Collaborating with medical institutions for clinical trials will provide comprehensive validation of our models, and involving ophthalmologists will be crucial for assessing accuracy and reliability in real-world scenarios. Optimizing the model for lower computational requirements and real-time processing will enhance its practical deployment in clinical settings.

In summary, this study introduces a robust strabismus detection system that addresses significant gaps in current literature. By incorporating both cross-eye and direct eye regression models, we have developed a method that offers superior accuracy and reliability. Although there are limitations to address, the findings lay a solid foundation for future research and practical applications, ultimately aiming to improve patient outcomes and reduce the burden on healthcare professionals.

## 4. Conclusions

In this study, we introduced an innovative approach to strabismus detection by integrating both direct and cross-eye regression models. Our best-performing configuration comprises the Xception model, the RMSprop optimizer, and specific training epochs for each model (325 epochs for cross-eye and 250 for direct eye). For real-life video analysis, we employed a classification threshold based on the average Euclidean distance plus 1.5 times the standard deviation of the normal eye dataset.

Our approach demonstrated excellent stability, consistency, and accuracy in estimating pupil positions for both normal and strabismus cases, making it suitable for clinical deployment. Importantly, the flexibility of our system allows it to be used in various screening scenarios without requiring subjects to sit at a fixed distance or maintain a rigid gaze. This adaptability enhances its practical use in real-world applications.

However, we encountered challenges during benchmarking, such as grayscale image formats, variations in eye regions, and image annotations. These issues highlighted the need for nuanced analysis and further model refinement. We also recognize the limitation of relying on a dataset that predominantly includes adult subjects, underscoring the need for pediatric-specific data and clinical validation.

Future research should focus on expanding the dataset to include a more diverse range of eye conditions and demographics, particularly younger populations. Collaborating with medical institutions to conduct clinical trials will provide comprehensive validation of our models. Involving ophthalmologists in assessing the accuracy and reliability of strabismus detection in real-world scenarios is essential. Additionally, optimizing the model for lower computational requirements and real-time processing will enhance its practical deployment in clinical settings.

Our study lays a solid foundation for future advancements, encouraging collaboration and knowledge sharing. We envision strabismus detection becoming integral to clinical practice, transforming early diagnosis and treatment monitoring. The integration of advanced image processing, extensive benchmarking, and AI techniques holds the potential to further enhance the accuracy and accessibility of strabismus detection, surpassing human capabilities in many scenarios.

In conclusion, our study presents a robust and flexible strabismus detection approach, paving the way for further exploration and innovation in this field. The integration of direct and cross-eye regression models reflects our commitment to advancing knowledge and technology for the benefit of individuals with strabismus, particularly in telemedicine contexts.

## Figures and Tables

**Figure 1 jimaging-10-00284-f001:**
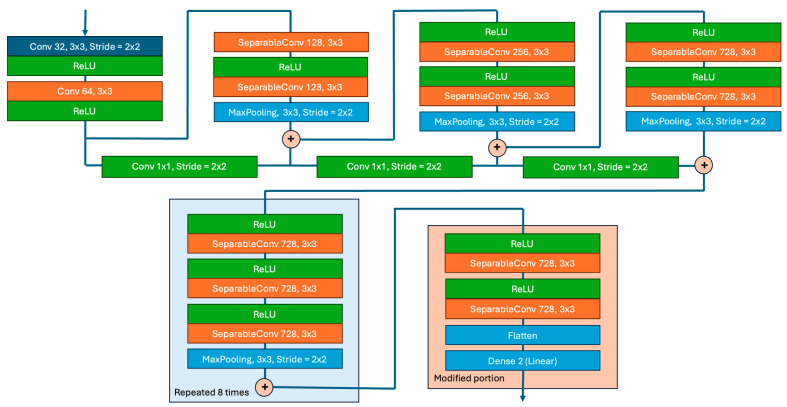
Diagram of the modified Xception architecture used to create cross-eye and direct eye regression models for strabismus detection, featuring regression layers in place of the classification head.

**Figure 2 jimaging-10-00284-f002:**
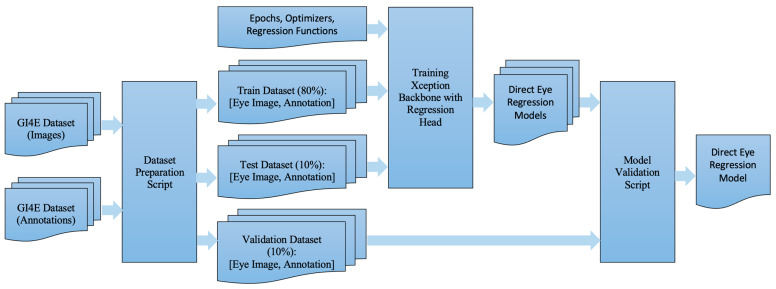
Summary of steps for direct eye regression model preparation.

**Figure 3 jimaging-10-00284-f003:**
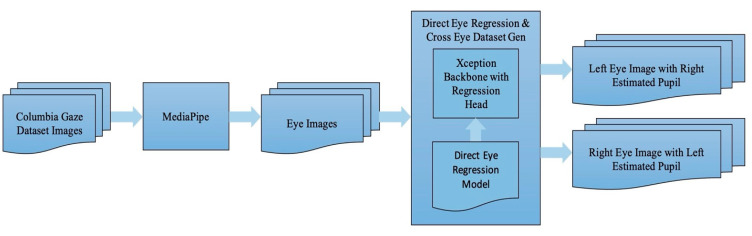
Summary of steps for generating datasets for cross-eye model training.

**Figure 4 jimaging-10-00284-f004:**
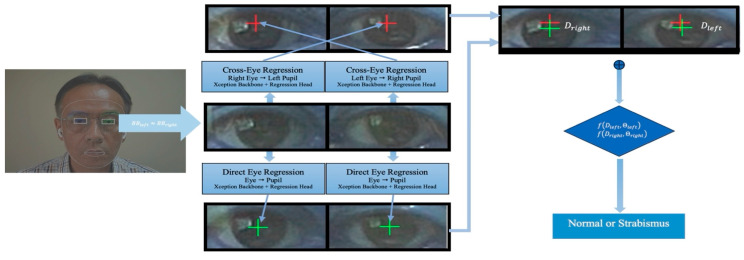
Overview of the proposed methodology for strabismus detection.

**Figure 5 jimaging-10-00284-f005:**
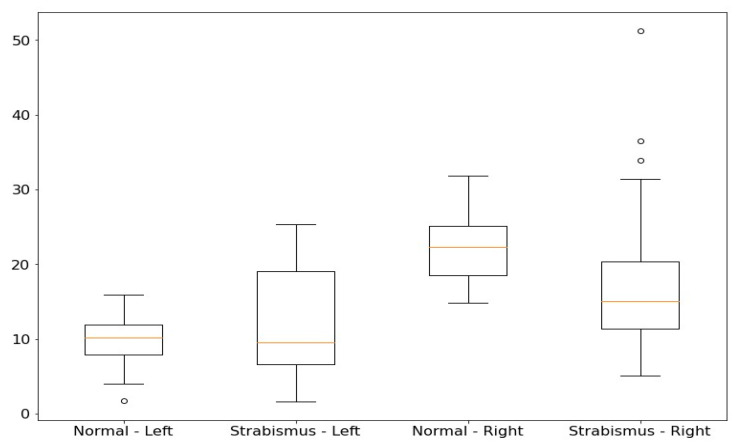
The box plot results from our comprehensive statistical analysis of the Huang et al. [3] dataset.

**Figure 6 jimaging-10-00284-f006:**
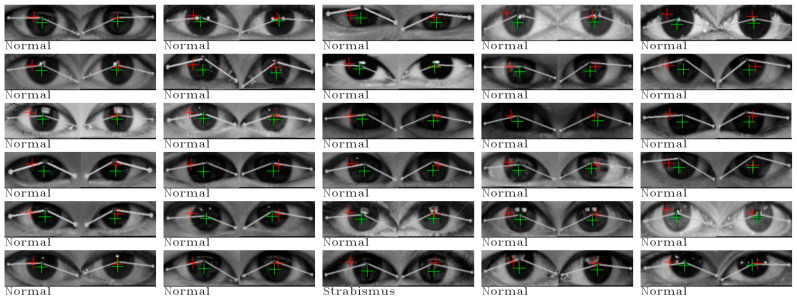
All classification results from our approach for normal eyes in the Huang et al. [3] dataset.

**Figure 7 jimaging-10-00284-f007:**
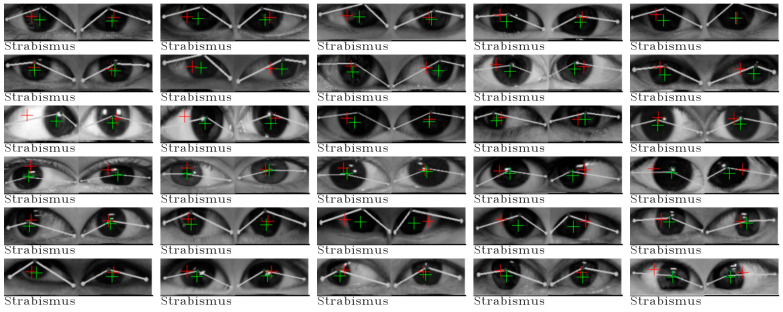
All classification results from our approach for strabismus in the Huang et al. [3] dataset.

**Figure 8 jimaging-10-00284-f008:**
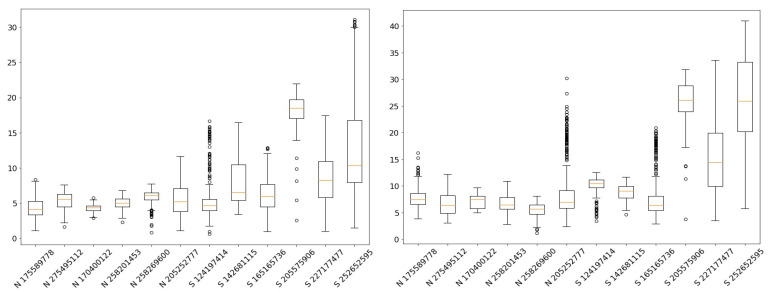
The box plot results from our comprehensive statistical analysis of the video dataset (N: Normal Eye and S: Strabismus).

**Table 1 jimaging-10-00284-t001:** Video datasets from Dreamstime: video IDs used as search keys for video clip retrieval.

Video ID	Eye Type	Total Duration (sec)	Duration (sec) and Coverage	Percent of Strabismus
175589778	Normal	12		
275495112	Normal	5		
170400122	Normal	6		
258201453	Normal	8		
258269600	Normal	18		
205252777	Normal	13		
124197414	Strabismus	10	4 [00:01–00:04]	40
142681115	Strabismus	7	5 [00:00–00:04]	71
165165736	Strabismus	9	3 [00:03–00:05]	33
205575906	Strabismus	6		100
227177477	Strabismus	32		100
252652595	Strabismus	33		100

**Table 2 jimaging-10-00284-t002:** Cross-eye regression Xception models results.

Model Configuration	Left Eye					Right Eye		
	*MSE*	SD	Min	Max	*MSE*	SD	Min	Max
Adam, MAPE	6.61				9.85			
RMSprop, MAE (250 Epochs)	6.07				9.47			
RMSprop, MAE (300 Epochs)	6.26	1.87	1.16	14.98	11.50	3.25	3.46	21.94
RMSprop, MAE (305 Epochs)	6.69	2.08	1.12	14.56	10.71	3.37	2.91	21.67
RMSprop, MAE (310 Epochs)	6.23	2.01	1.26	15.12	12.38	4.02	3.89	29.19
RMSprop, MAE (315 Epochs)	6.13	1.93	1.10	13.91	10.07	3.22	1.52	20.55
RMSprop, MAE (320 Epochs)	5.77	2.11	0.21	15.28	10.18	3.22	1.85	21.52
RMSprop, MAE (325 Epochs)	5.25	2.21	0.56	14.29	9.10	3.40	1.74	19.91
RMSprop, MAE (330 Epochs)	6.03	2.03	0.57	14.91	11.03	3.27	2.70	22.02
RMSprop, MAE (335 Epochs)	5.81	1.97	1.08	14.50	9.30	3.22	0.82	19.85
RMSprop, MAE (340 Epochs)	5.73	1.95	0.18	13.80	10.04	3.41	1.60	21.53
RMSprop, MAE (345 Epochs)	5.98	2.07	0.72	14.74	11.15	3.45	2.94	24.62
RMSprop, MAE (350 Epochs)	5.75	1.83	0.68	13.92	9.65	3.44	2.27	20.38
RMSprop, MAE (375 Epochs)	6.28	2.13	0.40	15.35	11.61	4.24	2.96	29.64
RMSprop, Log Cosh	6.51				10.01			
RMSprop, Huber Loss	5.89				11.01			
RMSprop, Mean Absolute Percentage Error	6.02				9.59			
RMSprop, Mean Absolute Percentage Error	6.47				11.40			
RMSprop, CosineSimilarity	13.00				13.66			

**Table 3 jimaging-10-00284-t003:** Direct eye Xception regression models results at 250 epochs.

Model Configuration		Metrics		
	MSE	SD	Min	Max
Nadam MAE	5.37	2.26	0.49	12.03
Adam Log Cosh	5.69	2.22	0.18	12.81
Adam MAE	5.82	2.35	0.04	12.64
Nadam MSE	6.03	2.09	0.42	12.12
Adam Mean Absolute Percentage Error	6.18	2.09	0.55	12.70

**Table 4 jimaging-10-00284-t004:** Benchmark results: our approach applied to Huang et al. [3] dataset.

Eye Type		Left Eye				Right Eye			Percent Correctness
	Mean	SD	Min	Max	Mean	SD	Min	Max	
Normal	9.66	3.11	1.79	15.95	21.97	4.09	14.89	31.82	96.67%
Strabismus	12.34	7.07	1.65	25.33	18.36	10.05	5.09	27.82	100%

**Table 5 jimaging-10-00284-t005:** Mean, standard deviation, minimum, and maximum Euclidean distances for left and right eyes in video clips.

Video ID	Frames %		Left Eye				Right Eye			Classifications		
		Mean	SD	Min	Max	Mean	SD	Min	Max	Left	Right	Final
175589778	75	5.70	1.99	0.78	10.72	9.08	2.48	3.49	17.48	N	N	N
275495112	89	5.36	1.24	1.64	7.63	6.57	1.94	3.08	12.18	N	N	N
170400122	49	4.24	0.59	2.86	5.72	7.11	1.29	4.96	9.65	N	N	N
258201453	51	5.02	0.84	2.31	6.78	6.64	1.60	2.80	10.93	N	N	N
258269600	96	5.92	0.83	0.86	7.73	5.54	1.19	1.13	8.08	N	N	N
205252777	43	5.57	2.21	1.09	11.66	9.34	5.59	2.41	30.23	N	N	N
µ		5.30				7.38						
σ		0.60				1.51						
µ + 1.5 × σ		6.21				9.64						
124197414	85	5.82	3.40	0.60	16.68	10.12	1.67	3.42	12.56	N	S	S
142681115	92	8.19	3.58	3.41	16.46	8.94	1.32	4.65	11.7	S	N	S
165165736	92	6.18	2.22	0.99	12.83	7.80	4.05	2.91	20.85	N	N	N
205575906	34	17.92	2.95	2.55	21.95	25.79	4.19	3.80	31.85	S	S	S
227177477	78	8.55	3.32	1.06	17.46	15.06	6.44	3.51	33.58	S	S	S
252652595	79	12.80	6.67	1.49	31.05	26.03	8.52	5.77	40.94	S	S	S

## Data Availability

Data available on request from the authors.

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
