# Peer review of "Strabismus Detection in Monocular Eye Images for Telemedicine Applications"

_2313-433X, 2024, doi:10.3390/jimaging10110284_

Round 1
Reviewer 1 Report
Comments and Suggestions for Authors
The manuscript makes a good impression and is worthy of publication after some corrections.
(see the rev.pdf)

Author Response
Dear Reviewer,
Thank you for your valuable feedback on our manuscript, "Strabismus Detection in Monocular Eye Images for Telemedicine Applications." We appreciate your insightful comments and suggestions.
In response to your observation regarding the mention of pediatric screening, we have revised the title and abstract to focus solely on the method's potential for telemedicine and early strabismus detection, without explicitly referencing pediatric applications. We have also included discussions about the possibility of pediatric screening in the discussion and conclusion sections.
Additionally, we have made modifications to ensure clarity and accuracy throughout the manuscript. Your input has significantly contributed to improving the quality of our work.
Thank you once again for your constructive feedback. We look forward to your thoughts on our revised manuscript.

Reviewer 2 Report
Comments and Suggestions for Authors
Using an eye toward its application in telemedicine and pediatric assessment, this research presents an original method for early identification of strabismus, a common ocular misalignment disorder. In non-strabismic cases, the technique uses coordinated eye motions to find the pupil position of one eye by reference to the other, therefore displaying near alignment. The technique presents a non-invasive, efficient approach for early detection, especially in telemedicine, serving as a dynamic screening tool for children without requiring patients to maintain a specific distance during assessment. Should those issues be addressed, this research possesses the possibility for publication in the MDPI Journal of Imaging:
1. In lines 123 and 124, it’s written that “It consists of images from 56 subjects...” Please provide some references that prove this subject is qualified for this work.
2. It’s better to put Figure 1 on page 5, it will make readers understand the explanation of Figure 1.
3. Please locate the title of all the figures in the center alignment.
4. In the datasets and methods section there are [21], [22], and [23] quoted but it’s cannot be found in the reference section.
5. For line 382, “Table 3. Direct Eye Xception Regression Models Results at 250 Epochs.” Be consistent with the text style.
6. For Figure 4, Figure 5, and Figure 6 in the title of the figures, please be consistent with the text style.
7. Please make Table 4 fit with the page.
8. Please unite Table 5, do not separate it.
9. Add space after Table 5.
10. In line 528, “The findings suggest that our dual regression method can effectively enhance the early diagnosis and monitoring of strabismus, offering a reliable tool for clinicians.” Please add some data that proves this finding can become a reliable tool for clinicians.
11. In line 547, “Future work should prioritize collecting and incorporating pediatric data to better tailor the model to younger populations.” It’s better to determine the ideal age range that can be used in this study.
12. Please move line 590, “Conclusion” to the next page.
13. In the reference section, for number 7 (Economides, J.R.; Adams, D.L.; Horton, J.C. Variability of Ocular Deviation in Strabismus), but I can not find it in the introduction section that refers to this number.
14. To create a comprehensive study of your study, it would be better combining this article with other recent studies such as fully forthcoming head-mounted displays [Photonics Research, Vol. 10, Issue 1, pp. 21–32 (2022)], holographic augmented reality [Laser & Photonic Reviews Vol. 16, Issue 6, 2100638 (2022)], and virtual content generation [Optics Letters Vol. 48, Issue 11, pp. 2809–2812 (2023)].
Author Response
Dear Reviewer,
Thank you for your detailed and constructive feedback on our manuscript. We have carefully reviewed each of your comments and have made the necessary revisions to enhance the quality and comprehensiveness of the study. To provide a clear overview of our responses, we have attached a PDF file that contains a detailed list of the comments, our corresponding responses, and the specific changes made to the manuscript.
The attached PDF includes:
- Revisions to the discussion section, incorporating suggested advancements like head-mounted displays, holographic augmented reality, and virtual content generation as potential directions for future research.
- Adjustments to the reference list and in-text citations, ensuring consistency and accuracy as per your suggestions.
- Modifications to figures, tables, and formatting, aligning with your recommendations to improve the presentation of the manuscript.
We hope that these revisions meet your expectations and contribute to the clarity and comprehensiveness of our study.
Thank you once again for your time and valuable feedback.

Round 2
Reviewer 2 Report
Comments and Suggestions for Authors
Although the authors claimed head-mounted displays, holographic augmented reality, and virtual content generation are expensive in the introdcution, the authors are required to have a performance comparison on thier work and the work involving the strabismus correction using head-mounted displays, holographic augmented reality, and virtual content generation in the revised manuscript.
Author Response
Comment 1: Although the authors claimed head-mounted displays, holographic augmented reality, and virtual content generation are expensive in the introduction, the authors are required to have a performance comparison on their work and the work involving the strabismus correction using head-mounted displays, holographic augmented reality, and virtual content generation in the revised manuscript.
Respond 1: Thank you for your insightful comment regarding the performance comparison of our work with head-mounted displays, holographic augmented reality, and virtual content generation technologies for strabismus correction. We appreciate the importance of such a comparison.
However, acquiring access to these advanced technologies for direct performance comparison poses significant challenges due to their cost, availability, and the specific expertise required to operate them. While we recognize the relevance of these technologies in the field, we believe that conducting a direct comparison within the current constraints may not be feasible.
In the revised manuscript, we have emphasized this limitation in the discussion section and suggested that future research could explore comparative studies as these technologies become more accessible. We also highlighted the unique strengths of our proposed method, particularly in terms of cost-effectiveness and practicality for widespread use in various healthcare settings. Additionally, we acknowledged the necessity of pediatric-specific data for improving the applicability of our model in younger populations, which is a critical area for future research.
We appreciate your understanding of this matter and believe that these amendments will enhance the manuscript's overall context and clarity.